# Learning Discrete Adaptive Receptive Fields for Graph Convolutional Networks

## Abstract

Different nodes in a graph neighborhood generally yield different importance. In previous work of Graph Convolutional Networks (GCNs), such differences are typically modeled with attention mechanisms. However, as we prove in our paper, soft attention weights suffer from over-smoothness in large neighborhoods. To address this weakness, we introduce a novel framework of conducting graph convolutions, where nodes are discretely selected among multi-hop neighborhoods to construct adaptive receptive fields (ARFs). ARFs enable GCNs to get rid of the over-smoothness of soft attention weights, as well as to efficiently explore long-distance dependencies in graphs. We further propose GRARF (GCN with Reinforced Adaptive Receptive Fields) as an instance, where an optimal policy of constructing ARFs is learned with reinforcement learning. GRARF achieves or matches state-of-the-art performances on public datasets from different domains. Our further analysis corroborates that GRARF is more robust than attention models against neighborhood noises.

## 1 Introduction

After a series of explorations and modifications (Bruna et al., 2014; Kipf & Welling, 2017; Velickovic et al., 2017; Xu et al., 2019; Li et al., 2019; Abu-El-Haija et al., 2019), Graph Convolutional Networks (GCNs) [1] have gained considerable attention in the machine learning community. Typically, a graph convolutional model can be abstracted as a message-passing process (Gilmer et al., 2017) – nodes in the neighborhood of a central node are regarded as contexts, who individually pass their messages to the central node via convolutional layers. The central node then weighs and transforms these messages. This process is recursively conducted as the depth of network increases. [2]

Neighborhood convolutions proved to be widely useful on various graph data. However, some inconveniences also exist in current GCNs. While different nodes may yield different importance in the neighborhood, early GCNs (Kipf & Welling, 2017; Hamilton et al., 2017) did not discriminate contexts in their receptive fields. These models either treated contexts equally, or used normalized edge weights as the weights of contexts. As a result, such implementations failed to capture *critical contexts* – contexts that pose greater influences on the central node, close friends among acquaintances, for example. Graph Attention Networks (GATs) (Velickovic et al., 2017) resolved this problem with attention mechanisms (Bahdanau et al., 2015; Vaswani et al., 2017). Soft attention weights were used to discriminate importance of contexts, which allowed the model to better focus on relevant contexts to make decisions. With impressive performances, GATs became widely used in later generations of GCNs including (Li et al., 2019; Liu et al., 2019).

However, we observe that using soft attention weights in hierarchical convolutions does not fully solve the problem. Firstly, we will show as Proposition 1 that under common conditions, soft attention weights almost surely approach 0 as the neighborhood sizes increase. This *smoothness* [3] hinders the discrimination of context importance in large neighborhoods. Secondly, we will show by experiments

---

[1] We use the name GCN for a class of deep learning approaches where information is convolved among graph neighborhoods, including but not limited to the vanilla GCN (Kipf & Welling, 2017).

[2] We use the term *contexts* to denote the neighbor nodes, and *receptive field* to denote the set of contexts that the convolutions refer to.

[3] The smoothness discussed in our paper is different to that in (Li et al., 2018), *i.e.* the phenomenon that representations of nodes converge in very deep GNNs.

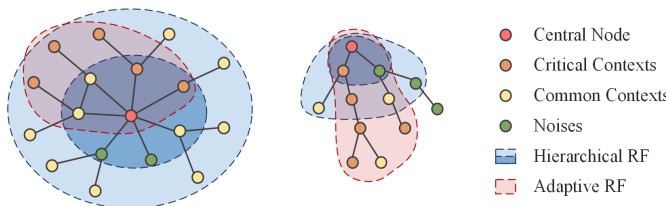

Figure 1: A Comparison between hierarchical convolutions and convolutions with ARF. Left: GCNs with ARFs better focus on critical nodes and filter out noises in large neighborhoods. Right: ARFs more efficiently explore long-distance dependencies.

in Section 4.2 that GATs cannot well distinguish true graph nodes from artificial noises: attention weights assigned to true nodes and noises are almost identical in distribution, which further leads to a dramatic drop of performance.

Meanwhile, an ideal GCN architecture is often expected to exploit information on nodes with various distances. Most existing GCNs use hierarchical convolutional layers, in which only one-hop neighborhoods are convolved. As a result, one must increase the model depth to detect long-distance dependencies (informative nodes that are distant from the central nodes). This is particularly an issue in large graphs, as the complexity of the graph convolutions is exponential to the model depth. [4] In large graphs, the model depths are often set as 1, 2 or 3 (Hamilton et al., 2017; Velickovic et al., 2017). Accordingly, no dependencies longer than 3 hops are exploited in these models.

Motivated by the discussions above, we propose the idea of **adaptive receptive fields (ARFs)**. Figure 1 illustrates the differences between hierarchical convolutions and convolutions with ARFs. An ARF is defined as a subset of contexts that are most informative for a central node, and is constructed via selecting contexts among the neighborhood. Nodes in an ARF can be at various distances from the central node. The discrete selection process of contexts gets rid of the undesired smoothness of soft weights (see Section 2). In addition, by allowing ARFs to choose contexts on different hops from the central node, one can efficiently explore dependencies with longer distances. Experiments also show that ARFs are more robust to noises (see Section 4). We further propose **GRARF (GCNs with Reinforced Adaptive Receptive Fields)** as an instance for using ARFs in node-level tasks. In GRARF, an optimal policy of constructing ARFs is learned with reinforcement learning (RL). An RL agent (*constructor*) successively expands the ARF via a two-stage process: a *contact* node in the intermediately-constructed ARF is firstly selected; a context among the direct neighbors of the contact node is then added to the ARF. The reward of the constructor is defined as the performance of a trained GCN (*evaluator*) on the constructed ARF.

GRARF is validated on datasets from different domains including three citation networks, one social network, and an inductive protein-protein interaction dataset. GRARF matches or improves performances on node classification tasks compared with strong baselines. [5] Moreover, we design two tasks to test the models' abilities in focusing on informative contexts and leveraging long-distance dependencies by injecting node noises in graphs with different strategies.

## 2  PRELIMINARIES AND THEORIES

**Notations.** In our paper, we consider node-level supervised learning tasks on attributed graphs. An attributed graph $G$ is generally represented as $G = (V, A, X)$, where $V = \{v_1, \cdots, v_n\}$ denotes the set of nodes, $A \in \{0, 1\}^{n \times n}$ denotes the (binary) adjacency matrix, and $X \in \mathbf{R}^{n \times d_0}$ denotes the input node features, $x_v \in \mathbf{R}^{d_0}$ the features of node $v$. $E$ is used as the set of edges. We use $N(v_i)$ to denote the one-hop neighborhood of node $v_i$, with $v_i$ itself included. We use $H^{(l)} \in \mathbf{R}^{n \times d_l}$ as the matrix containing $d_l$-dimensional hidden representations of nodes in the $l$-th layer, $h_v^{(l)}$ that of node

---

[4]With sparse adjacency matrices, the average complexity of graph convolutions is $O(d^L)$, where $L$ is the model depth and $d$ is the graph degree (or the neighborhood-sampling sizes in (Hamilton et al., 2017)).

[5]We mainly show the results of node classification tasks in our paper, whereas GRARF is intrinsically adapted to all node-level supervised learning tasks.

$v$. $\hat{A}$ denotes the symmetrically normalized adjacency matrix with $\hat{A} = D^{-1/2}(A + I_n)D^{-1/2}$ and $D = \text{diag}(d), d_i = \sum_j (A + I_n)_{ij}$. We use bold letters for neural network parameters.

**The smoothness of Graph Attention Networks.** As a pioneering work of simplifying architectures of graph neural networks, the vanilla GCN layers in (Kipf & Welling, 2017) were defined as

$$H^{(l+1)} = \sigma\left(\hat{A}H^{(l)}\mathbf{W}^{(l)}\right) = \sigma\left(\sum_{j \in N(v_i)} \hat{A}_{ij} h_j^{(l)} \mathbf{W}^{(l)}\right), \quad l = 0, 1, \cdots \tag{1}$$

In each layer, the node representations in one-hop neighborhoods were transformed with $\mathbf{W}$ and averaged by normalized edge weights $\hat{A}_{ij}$. Graph Attention Networks (GATs) (Velickovic et al., 2017) elaborated the average scheme in Eq. (1) with attention mechanisms (Bahdanau et al., 2015; Vaswani et al., 2017). Instead of using $\hat{A}_{ij}$, an attention weight $\alpha_{ij}$ between node $v_i$ and $v_j$ was calculated in GAT layers as

$$e_{ij} = f_\theta(h_i, h_j), \quad \alpha_{ij} = \text{softmax}_j(e_{ij}) = \frac{\exp(e_{ij})}{\sum_{k \in N(v_i)} \exp(e_{ik})}, \tag{2}$$

where $f_\theta(\cdot)$ is often called the *energy function* with parameter $\theta$. GATs implicitly enabled specifying different weights in a neighborhood. However, under some common assumptions and as the neighborhood size increases, these attention weights, normalized with the *softmax* function, suffer from over-smoothness: all attention weights approach 0 as the neighborhood size increases. We formally introduce and prove this claim as Lemma 1 and Proposition 1:

**Lemma 1** (the smoothness of softmax). *If random variables $X_1, X_2, \cdots$ are uniformly bounded with probability 1, that is, for any $i$ and some $C$, $P(|X_i| > C) = 0$, then the softmax values taking $\{X_i\}_{i=1}^n$ as inputs approach 0 almost surely when $n \to \infty$, i.e.*

$$e^{X_i} \Big/ \sum_{j=1}^n e^{X_j} \to 0 \quad a.s. \tag{3}$$

*Proof.* The proof is simple noting that $e^{X_i}/\sum_{j=1}^n e^{X_j} > 0$, and that with probability 1,

$$e^{X_i} \Big/ \sum_{j=1}^n e^{X_j} < e^C \Big/ ne^{-C} \to 0, \quad n \to \infty. \quad \square$$

**Proposition 1** (the smoothness of attention weights). *If the representation of nodes (random vectors) $H_1, H_2, \cdots \in \mathbf{R}^d$ are uniformly bounded with probability 1 (for any $i$ and some $C$, $P(\|H_1\| > C) = 0$), and for any (fixed) node $v_i$, the energy function $f_\theta(h_i, \cdot)$ is continuous on any closed set $D \in \mathbf{R}^d$, then the attention weights in the neighborhood of $v_i$ approach 0 almost surely when $n \to \infty$, i.e.*

$$\alpha_{ij} = \frac{\exp(f_\theta(h_i, H_j))}{\sum_{k \in N(v_i)} \exp(f_\theta(h_i, H_k))} \to 0 \quad a.s. \tag{4}$$

*Proof.* Following the *a.s.* boundedness $\{H_i\}$s and the continuity condition on $f_\theta(\cdot)$, the random energies $E_{ij} = f_\theta(h_i, H_j)$ are also bounded *a.s.*. The desired result then follows Lemma 1. $\square$

Note that the continuity condition on $f_\theta(h_i, \cdot)$ in Proposition 1 can be satisfied with almost any commonly used non-linear functions and (regularized) parameters in deep learning, specifically, those in the official version of GATs ($e_{ij} = a^T[\mathbf{W}h_i \| \mathbf{W}h_j]$, where $\|$ is the operator of concatenation). Also, the boundedness of inputs is trivial in deep learning.

**GCNs with ARFs overcome smoothness.** What Proposition 1 shows is that in large neighborhoods, attention weights are smoothed to 0, thus hindering the discrimination of context importance. In addition, such smoothness can be immediately generated to any other form of normalized weights as long as $\alpha_{ij} > 0$ uniformly and $\sum_{j \in N(v_i)} \alpha_{ij} = 1$. We alleviate the smoothness with ARFs by incorporating discreteness. Specifically, let us denote the convolution in the evaluator as

$$h_i' = \sigma\left(\sum_{j \in N_a(u)} \eta_{ij} h_j W\right) = \sigma\left(\sum_{j \in N^k(u)} \tilde{\eta}_{ij} h_j W\right), \quad \tilde{\eta}_{ij} = \begin{cases} \eta_{ij}, & j \in N_a(u), \\ 0, & j \notin N_a(u), \end{cases} \tag{5}$$

where $N_a(u)$ is an ARF, $k$ is the maximum hop that the ARF explores, and $N^k$ is the entire $k$-hop neighborhood. Accordingly, $\tilde{\eta}_{ij}$ is not subjected to smoothness: if $\eta_{ij}$ has a uniform lower bound $D > 0$, then for $p \in N_a(u)$ and $q \notin N_a(u)$, we have $\tilde{\eta}_{ip} - \tilde{\eta}_{iq} > D$, regardless of the sizes of $N^{(k)}$. Note that $\eta_{ij} > 0$ uniformly can be guaranteed in most cases when the maximum ARF size is limited, for example, with uniform weights or softmax weights of bounded energies.

It should be noted that the discrimination of nodes in ARFs is different to that in MixHop (Abu-El-Haija et al., 2019), which directly takes multi-hop nodes as inputs. MixHop can specify different *parameters* [6] for contexts *on different hops* (*hop-level* discrimination), while it CANNOT specify different *weights* for contexts *on the same hop* (*node-level* discrimination) as they were uniformly treated (averaged). The two levels of discrimination are orthogonal, and we focus on the latter.

**Deep reinforcement learning on graphs.** As the discrete context selection process is non-differentiable, we apply deep reinforcement learning approaches to learn the policy of constructing ARFs in GRARF, specifically, the Deep Q-Learning (DQN) (Mnih et al., 2015) algorithm. DQN uses deep neural networks to approximate the *action value function* (*Q-function*), and chooses the action that maximizes it in each step. The Q-function is defined iteratively as

$$Q^*(s_t, a_t) = R(s_t, a_t, s_{t+1}) + \gamma \max_{a \in \mathcal{A}} Q^*(s_{t+1}, a), \tag{6}$$

where $s$ is the state, $R(\cdot)$ is the reward function, $\mathcal{A}$ is the action space, and $\gamma$ is a discount factor. A *reward shaping* technique (Ng et al., 1999) is also used in GRARF to alleviate the sparsity of rewards, which decorates the original reward $R(\cdot)$ with a *potential energy* $F(\cdot)$, yielding an *immediate reward* $\hat{R}(\cdot)$. Denoted in formula,

$$F(s, a, s') = \Phi(s') - \Phi(s), \quad \hat{R}(s, a, s') = R(s, a, s') + F(s, a, s'), \tag{7}$$

where $\Phi(\cdot)$ is a fixed *potential function* of states that does not change during training. (Ng et al., 1999) proved that the optimal policies of MDPs remain invariant if $R(\cdot)$ is replaced by $\hat{R}(\cdot)$.

There are other recent papers implementing reinforcement learning on graphs. For example, GCPN (You et al., 2018) proposed an RL agent for generating graph representations of biomedical molecules, and DGN (Jiang et al., 2020) introduced a multi-agent reinforcement learning approach where the agents in the system formed a dynamic network. The successive molecule generation process in GCPN inspired us in designing the ARF constructor in GRARF, whereas the two models are of different motivations and applications.

**ARFs and neighborhood sampling.** It should be noted that GRARF can also be interpreted as a neighborhood sampling approach. Neighborhood sampling was proposed as a necessary process to apply GCNs to large graphs with arbitrarily large neighborhoods. GraphSAGE (Hamilton et al., 2017) proposed a general framework of neighborhood sampling and aggregation, where contexts were uniformly sampled. Later work improved the sampling strategy with importance sampling (Chen et al., 2018) and explicit variance reduction (Huang et al., 2018; Hamilton et al., 2017). Sub-graphs instead of subsets of neighborhoods were directly sampled in (Zeng et al., 2020). Indeed, selecting ARF nodes takes a specific form of neighborhood sampling. However, the aim of constructing ARFs is to *ignore* trivial information and to *focus* on critical contexts, rather than to *estimate* the neighborhood average as is the primary target of neighborhood sampling. Therefore, despite the similarity, the two approaches are in different directions.

## 3 PROPOSED METHOD: GRARF

### 3.1 ARF CONSTRUCTION AS MARKOV DECISION PROCESS

An adaptive receptive field is defined as a set of nodes $N_a(u)$ with regard to a central node $u$. Nodes in $N_a(u)$ can be at various distances from $u$, and $u$ itself should be contained in its ARF. We also assume that the *induced subgraph* of an ARF must be (weakly) connected, under the motivation that if a far context poses great influence on the central node, then at least one path connecting it to the central node should be included in the ARF. The ARF construction process is modeled as a Markov

---

[6]For clarity, we use the term *parameters* for parameters in the transformation matrix $W$ (see Eq.(1) in the original paper), and *weights* for the weights of different contexts (e.g. the attention weights).

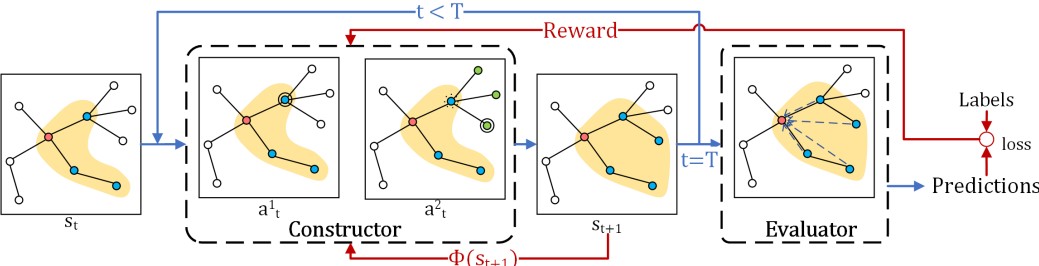

Figure 2: The general architecture of GRARF. Red and blue arrows together demonstrate the training process. Blue arrows along demonstrate the prediction process.

Decision Process (MDP) $M = (\mathcal{S}, \mathcal{A}, P, R, \gamma)$, where $\mathcal{S} = \{s_i\}$ is the state space of all possible ARFs; $\mathcal{A} = \{a_i\}$ describes all possible (two-stage) actions $a = (a^1, a^2)$; $P$ is the transition dynamics $p(s_{t+1}|s_t, a_t)$ which describes how nodes are added to ARFs; $R$ is the reward function of an ARF, and $\gamma$ is the discount factor.

Figure 2 introduces the general structure of our model. GRARF is composed of a *constructor* and an *evaluator*. The constructor implements an RL agent to learn an optimal policy of the MDP with DQN, and the evaluator conducts graph convolutions on constructed ARFs. Specifically, in the training phase, the constructor and the evaluator are trained alternately, where rewards of the constructor are derived from the performances of the evaluator; in the prediction phase, the evaluator convolves over the ARFs constructed for the target node by the constructor and then predicts the result. No gradient flows between the constructor and the evaluator.

## 3.2 MODEL ARCHITECTURE

We hereby specify the MDP of ARF construction, the evaluator, and the training scheme.

**States.** At step $t$, the state $s_t$ is defined as the intermediately generated ARF, which is encoded as a vector representation $\vec{s_t}$ by a function $f_s$, and then observed by the constructor. Formally,

$$s_t \triangleq N_a^{(t)}(u), \quad \vec{s_t} \triangleq f_s(N_a^{(t)}(u)), \tag{8}$$

We initialize the ARF as $N_a^{(0)}(u) = \{u\}$. The transition dynamics in GRARF is deterministic: nodes selected by the RL agent is added to the ARF, i.e. $N_a^{(t+1)}(u) = N_a^{(t)}(u) \cup a_t^2$.

**Actions.** For each action, the constructor chooses a node to add to the ARF among all nodes adjacent to the ARF. The average complexity of directly choosing among all adjacent candidates is $O(n_t \times d)$, where $n_t$ is the ARF size at step $t$ and $d$ is the graph degree. We reduce the complexity to $O(n_t + d)$ by decomposing the action into two stages, denoted as $a_t = (a_t^1, a_t^2)$. In the first stage, the constructor chooses a *contact node* $a_t^1 \in N_a^{(t)}(u)$, who limits the candidates of the next stage in its own direct neighborhood. In the second stage, the constructor searches among the neighborhood of $a_t^1$ for a node $a_t^2$ to add to the ARF. The optimal Q-function can accordingly be rewritten as

$$Q_1^*(s_t, a_t^1) = \max_{a_2 \in N(a_t^1)} Q_2^*(s_t, a_2), \tag{9}$$

$$Q_2^*(s_t, a_t^2; a_t^1) = R(s_t, a_t^2, s_{t+1}) + \gamma \max_{a_1 \in N_a^{(t+1)}(u)} Q_1^*(s_{t+1}, a_1). \tag{10}$$

We do not design explicit stop actions, and the process stops after a fixed number of steps ($T$). Meanwhile, we do not require that $a_t^2 \notin N_a^{(t)}(u)$, so a node may be selected multiple times in an ARF. Note that if a node already in the ARF is selected, the state ceases to change, and no new nodes will be selected. Therefore, various ARF sizes are implicitly allowed. The actions (i.e. candidate nodes) in both stages are encoded with $f_a$, and the approximated Q-function in GRARF is parameterized as

$$Q_1(s_t, a_t^1) = \mathbf{w}_1^T \left[ f_a(a_t^1) \| f_s(s_t) \right] + \mathbf{b}_1, \tag{11}$$

$$Q_2(s_t, a_t^2; a_t^1) = \mathbf{w}_2^T \left[ f_a(a_t^1) \| f_a(a_t^2) \| f_s(s_t) \right] + \mathbf{b}_2. \tag{12}$$

**Rewards.** The reward of the constructor is defined as the performance of the GCN evaluator. As conducting step-wise evaluations is much time-costly, we sample the reward once the ARF is fully constructed, that is,

$$R(s_t, a_t^2, s_{t+1}) \triangleq \begin{cases} 0, & t < T, \\ \text{eval}\left(N_a^{(t)}(u)\right) = -\text{loss}\left(\text{GCN}\left(N_a^{(t)}(u)\right)\right), & t = T. \end{cases} \quad (13)$$

The loss in Eq. (13) is specified by the downstream supervised node-level (or node-pair-level) tasks, such as node classification, link prediction, etc. We further adopt the *reward shaping* technique (Ng et al., 1999) to guide and accelerate the training process. Considering the desired properties of critical contexts and ARFs, we propose the following heuristic potential functions: i) $\Phi_1(s) = |N_a^{(t)}(u)|$, the sizes of ARFs, to encourage the variety of contexts; ii) $\Phi_2(s) = \sum_{v \in N_a(u)} \deg(v)$, the sum of degrees in the ARF, as nodes with higher degrees are intuitively more informative; iii) $\Phi_3(s) = \sum_{v \in N_a(u)} \text{sim}(v, u) = \sum_{v \in N_a(u)} x_v \cdot x_u$, the inner product of input features between the central node and the ARF nodes, to empirically encourage more relevant contexts. According to Eq. (7) with so-defined $\Phi$s, the immediate rewards of the constructor is

$$\hat{R}(s_t, a_t^2, s_{t+1}) = \begin{cases} 0, & t < T \quad \text{and} \quad a_t^2 \in N_a^{(t)}(u), \\ 1 + \deg(a_t^2) + x_{a_t^2} \cdot x_u, & t < T \quad \text{and} \quad a_t^2 \notin N_a^{(t)}(u), \\ -\text{loss}\left(\text{GCN}\left(N_a^{(t)}(u)\right)\right), & t = T. \end{cases} \quad (14)$$

**Evaluator.** Given a central node $u$ and the constructed ARF $N_a(u)$, the evaluator takes the ARF as neighborhood and convolves over it, generating the representation of the central node. An example of evaluator would be the one defined in Eq. (5). The representation is then used to conduct downstream tasks. Theoretically, any graph convolutional layers can be used as the evaluator. It is worth noting that although in our experiments we only perform a one-layer graph convolution on the constructed ARF, multiple convolutional layers can be applied on the subgraph induced by the ARF node set. We leave this as future work.

**Training.** In order to mutually train the constructor and the evaluator in GRARF, we propose an *alternate training strategy*, somehow analogous to the training of GAN (Goodfellow et al., 2014). Specifically, the evaluator is first pre-trained for the given downstream task, taking direct neighborhoods as receptive fields. We then fix the evaluator to derive constant task-aware rewards for the training of the constructor. The alternate process goes recursively until convergence. In details, as the training of the evaluator converges much faster than the constructor, we train the constructor with more steps in the alternate process. Empirically, 10 : 1 is a promising choice of the ratio.

## 4 EXPERIMENTS

### 4.1 EXPERIMENT SETUP

**Datasets.** We evaluate GRARF on public real-world datasets including 3 citation networks, a social network, and a protein-protein interaction dataset. Some interesting statistics of the datasets are shown in Table 1. In the citation networks (`cora`, `citeseer` and `pubmed`), [7] nodes correspond to publications in disjoint fields, and edges to (undirected) citation relationships. The social network (`github`) [8] consists of website users (nodes) and their friendships (edges). We reduce the number of input features of social networks to the figures in Table 1 by selecting most frequent ones (all features binary and sparse). The protein-protein interaction (`ppi`) dataset[9] contains 24 graphs, each representing a human tissue, where nodes denote different proteins and edges denotes the interactions in between. We use the preprocessed data of GraphSAGE (Hamilton et al., 2017).

**Baselines.** We pick up 5 GCN baselines to compare GRARF against, including vanilla GCN (Kipf & Welling, 2017), GraphSAGE (Hamilton et al., 2017), GAT (Velickovic et al., 2017), GIN (Xu et al., 2019) and MixHop (Abu-El-Haija et al., 2019). For fairness, the dimensions of all hidden

---

[7] Available at `https://linqs.soe.ucsc.edu/data`.

[8] Available at `http://snap.stanford.edu/data/`.

[9] Available at `http://snap.stanford.edu/graphsage/ppi.zip`.

Table 1: Interesting statistics of the datasets used in this paper. *(\*: multi-label task)*

|  | cora | citeseer | pubmed | github | ppi* |
|---|---|---|---|---|---|
| # Nodes | 2,708 | 3,327 | 19,717 | 37,700 | 56,944 |
| # Links | 5,429 | 4,732 | 44,338 | 289,003 | 818,716 |
| # Classes | 7 | 6 | 3 | 2 | 121* |
| # Features | 1,433 | 3,703 | 500 | 600 | 50 |
| Avg. Degree | 2.00 | 1.42 | 2.25 | 7.67 | 28.8 |

Table 2: Performances of GRARF and baselines on node classification tasks. *(% micro-f1)*

|  | cora | citeseer | pubmed | github | ppi |
|---|---|---|---|---|---|
| Vanilla GCN (Kipf & Welling, 2017) | $87.40 \pm 0.36$ | $78.09 \pm 0.35$ | $86.18 \pm 0.10$ | $81.72 \pm 0.14$ | $59.26 \pm 0.03$ |
| GraphSAGE (Hamilton et al., 2017) | $82.02 \pm 0.31$ | $70.23 \pm 0.51$ | $86.26 \pm 0.33$ | $78.09 \pm 1.23$ | $56.97 \pm 0.94$ |
| GAT (Velickovic et al., 2017) | $87.80 \pm 0.11$ | $76.43 \pm 0.73$ | $85.39 \pm 0.12$ | $81.78 \pm 0.52$ | $47.29 \pm 2.30$ |
| GIN (Xu et al., 2019) | $85.08 \pm 0.83$ | $73.87 \pm 0.52$ | $84.88 \pm 0.17$ | $79.05 \pm 0.88$ | $60.35 \pm 1.14$ |
| MixHop (Abu-El-Haija et al., 2019) | $\mathbf{88.70 \pm 0.19}$ | $74.59 \pm 0.27$ | $85.60 \pm 0.10$ | $79.68 \pm 1.00$ | $58.55 \pm 0.14$ |
| Random RF | $86.93 \pm 1.22$ | $77.31 \pm 1.37$ | $86.87 \pm 1.30$ | $79.81 \pm 7.82$ | $63.27 \pm 2.38$ |
| GRARF | $88.34 \pm 0.82$ | $\mathbf{79.24 \pm 0.88}$ | $\mathbf{88.20 \pm 0.33}$ | $\mathbf{87.30 \pm 2.54}$ | $\mathbf{66.54 \pm 1.12}$ |

representations are set as $d = 128$ and an identical two-layer setup is used in all baselines. Hyper-parameters such as learning rates in all models are tuned to achieve best performances on the validation sets. All neural models are trained with adequate epochs with the early-stopping strategy. More details of baselines are included in the Appendix.

**GRARF implementation.** In the constructor, all graph nodes are first encoded with a Graph-SAGE layer ($h_u^c = \text{elu}(\mathbf{W}_c[x_u \| \frac{1}{n} \sum_{v \in N(u)} x_v] + \mathbf{b}_c)$). For the state encoder $f_s$, we use linear transformations of concatenations of central node representations and ARF-averaged node representations, i.e. $f_s(N_a(u)) = \mathbf{W}_s[h_u^c \| \frac{1}{n} \sum_{v \in N_a(u)} h_v^c] + \mathbf{b}_s$. For the action encoder $f_a$, we directly use the hidden representations of nodes, i.e. $f_a(v) = h_v^c$. The discount factor is chosen as $\gamma = 0.9$. In the evaluator, a simple GraphSAGE layer convolves over constructed ARFs as $h_u^e = \text{elu}(\mathbf{W}_e[x_u \| \frac{1}{n} \sum_{v \in N_a(u)} x_v] + \mathbf{b}_e)$. The hidden representations $h_u^e$ are later used in node-level tasks with only one fully-connected layer. The same as all baselines, dimensions of all hidden representations in GRARF ($h_u^c$, $h_u^e$ and $\vec{s_t}$) are 128. To demonstrate the effectiveness of ARFs, we also conduct experiments with a random baseline, *Random RF*, where actions are chosen completely randomly, but strictly following the setup of GRARF.

## 4.2 RESULTS

**Performances of node classification tasks.** Table 2 shows the *micro-f1*s of GRARF and baselines on the node classification tasks including means and standard deviations across 10 replicas. The datasets are uniformly split to $5 : 2 : 3$ as training, validation and test sets, except for ppi, where 20 graphs are used as training set, 2 as validation set and 2 as test set, identical to that in (Hamilton et al., 2017). Tasks are transductive on citation and social network and inductive on ppi. GRARF shows very competitive performances on both types of tasks: on citation graphs where neighborhoods are small (degree $\approx 2$), GRARF matches or improves by margin the performances of other baselines; on github and ppi datasets with larger neighborhoods (degree $> 7$), GRARF displays a significant advantages over other baselines. This is in concordance with our expectations, as we have shown that GRARF overcomes the smoothness and thus better focuses on informative contexts in larger neighborhoods.

**Capturing informative contexts and long-distance dependencies (LDD).** To demonstrate the benefits of using ARFs, we design two experiments through adding artifical noises in cora with different strategies. **In the *denoising* setup**, we generate $r \times |V|$ noise nodes and randomly connects them to true nodes. An amount of $r \times |E|$ edges between true nodes and noises are added, so that the proportion of noises in the neighborhood is approximately $\frac{r}{1+r}$. **In the *LDD* setup**, we assign a *split number* $k$ drawn from $Poisson(\lambda)$ on each edge, and then split each edge to a $(k + 1)$-hop path (do nothing if $k = 0$) by inserting $k$ noises. The one-hop dependencies are hence stretched longer and more difficult for the models to detect. In both experiments, features of noises are drawn from marginal distributions of individual dimensions. We conduct experiments with 5 replicas per model and report the averaged performances and 95% confidence intervals. Figure 3 shows examples and performances of GRARF, vanilla GCNs under two setups. We also show the performances of raw-feature baseline for comparison (which fully ignores the noises).

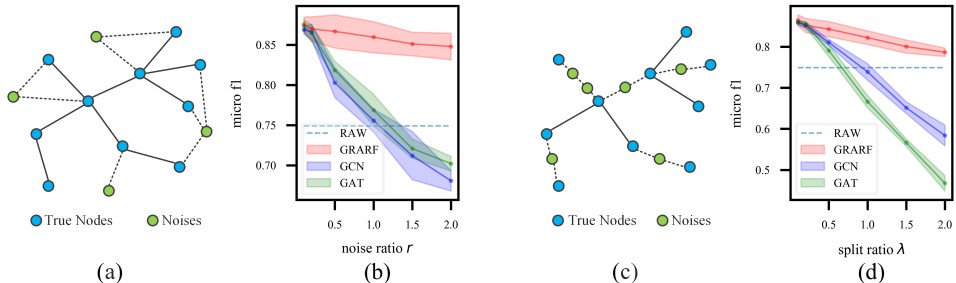

Figure 3: Examples and results of two designed experiments. (a)(b) The denoising experiment. (c)(d) The long-distance dependency experiment. GRARF displays robustness to noises in both experiments, whereas performances of vanilla GCNs and GATs drop dramatically.

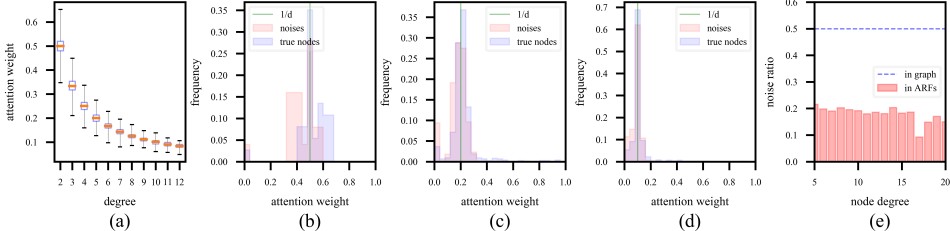

Figure 4: Analysis of behaviors in GAT and GRARF. (a) A box-plot of attention weights on `cora` with regard to node degrees, where medians, Q1s, Q3s and a $95\%$ intervals are displayed. (b)-(d) Histograms of attention weights assigned to true nodes and noises in the denoising experiments ($r = 1$), with different node degrees ($d = 2, 5, 10$). (e) The ratios of noises in constructed ARFs with regard to node degrees, in the denoising experiments ($r = 1$).

In the *denoising* experiment, GRARF displays a significantly better ability in capturing informative nodes while ignoring noises. The performances of vanilla GCNs and GATs drop dramatically as $r$ increases and even below the raw baseline when $r > 1$, whereas the performances of GRARF drops only marginally. A similar phenomenon is observed in the *LDD* experiment, which corroborates that the constructed ARFs in GRARF better collect the information from far-hop neighbors through exploring the neighborhood in a more flexible manner.

We further analyzed the behavior of GATs and GRARF on real and noisy data. Figure 4 (a) shows the distributions of attention weights that GATs assign to neighbors of central nodes (with degree $d$s) on original `cora`. The majority of attention weights lie in a very thin interval around $1/d$ which continues to shrink as $d$ increases. This is empirical evidence of Proposition 1. Figure 4 (b)-(d) show distributions of attention weights assigned to true nodes and noises in the *denoising* experiment with $r = 1$ and $d = 2, 5, 10$ (calculated in true nodes' neighborhoods only.). Attention weights assigned to true nodes and noises are almost identical, especially with larger $d$. This indicates that GATs cannot well distinguish noises from true nodes with attention weights. We also report the ratios of noises in constructed ARFs in the same experiment. The noise ratios in the ARFs stay far below the noise ratio in the graph, and remain almost invariant to the sizes of neighborhoods. This suggests that the ARF constructor learns to ignore noises in the graph neighborhoods.

## 5 CONCLUSION & FUTURE WORK

In this paper, we proposed the idea of adaptive receptive fields and GRARF as an instance. We showed both theoretically and empirically that GCNs with ARFs address the smoothness of attention weights, and hence better focus on critical contexts instead of common ones or noises. Meanwhile, as nodes in ARFs can be at various distances from the central node, GCNs with ARFs explore long-distance dependencies more efficiently. Nevertheless, the ARFs in our paper are simplified as sets of nodes, whose structures are mostly ignored.

For future work, a straight direction is to learn an optimal convolutional structures on constructed ARFs, or to jointly learn both. Finding fast approximations of RL is also attracting. Another promising prospect is to provide the constructor with more global-level information, for currently constructor only observes local neighborhoods.

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
