# OpenReview forum: "Learning Discrete Adaptive Receptive Fields for Graph Convolutional Networks"
_ICLR.cc/2021/Conference — Reject_

### Official Review · AnonReviewer4 · 2020-10-24
**Review - AnonReviewer4**

**Rating:** 5
**Confidence:** 4

**Review:**



**Summary:**

The paper proposes a method for avoiding the oversmoothing happening in standard GNN methods. It defines a receptive field of a node, as the set of nodes that send messages to that node and proposes a method to create adaptive receptive fields specific to each node. Instead of using all the nodes in a multi-hop neighbourhood, an reinforcement learning method is proposed to select only a subset of these nodes. The RL problem is formalised such that each state represents a set of already selected nodes, while actions represent the next selected node. The goal of the RL agent (constructor) is to form an adaptive neighbourhood for each node, and the reward is given by the loss of a GNN method (evaluator) that uses this neighbourhood.

**Strengths:**


Adapting the receptive field of each node in a GNN is a good idea that could result in better modeling of the context, longer connections and also avoiding feature oversmoothing.

The proposed RL method to create adaptive receptive fields is sound and experiments show that it is indeed useful.


The experiments with noisy data are interesting and show that ARF selects important nodes and also helps for long-range connections.



**Weaknesses**

The main weakness of the paper is that the proposed adaptive receptive field (ARF) is not properly motivated by the theory in section 2. It is a good idea to have an ARF as it has many advantages, but they are not properly explained in the theoretical section. ARF could achieve long-range connections while avoiding the standard node features oversmoothing problem (different than the presented attention smoothing), but this should be better motivated.


The smoothness of the weights should be more clearly defined. For example, are the weights $1 / n, 2 / n, 3 / n$ smooth? They respect Proposition 1 as they all approach 0 when $n \to \inf$, but they could be distinguished. Different from the usage in the paper, maybe the ratio $\alpha_{i,j} / \alpha_{i,k}, \forall j,k \in N(i)$ could be used.

It is not clear why small attention weights should be avoided. Weights that decrease with the number of nodes do not necessarily result in bad representations. All the attention weights could approach 0, but they could be sufficiently distinct and also result in proper node representations.

The computational complexity of the constructor should be discussed. What is the total inference time of GRARF, taking into account the construction of the adaptive fields? What percent of the total time is spent by the constructor and evaluator?





**Additional comments and questions:**

Fair evaluation on cora, citeseer, pubmed datasets has been shown to be difficult [A]. The community seems to be moving forward to larger datasets like OGB [B] or [C].

It would be useful to make a connection and compare to other approaches that involve adaptive receptive fields, like Liu et al., 2019.

The weight's smoothness should have a clearly presented definition, especially to avoid confusions to the smoothness of the node features, as pointed out by the authors in footnote 3.

How important is the reward shaping for good performance?

*Minor*:
In Eq.1 the first part represents the whole graph H, while the last term represents a single node:  h_i.
Eq. 5: u == i ?


[A] Shchur, Oleksandr, et al. "Pitfalls of graph neural network evaluation." arXiv preprint arXiv:1811.05868 (2018).
[B] Hu, Weihua, et al. "Open graph benchmark: Datasets for machine learning on graphs." arXiv preprint arXiv:2005.00687 (2020).
[C] Dwivedi, Vijay Prakash, et al. "Benchmarking graph neural networks." arXiv preprint arXiv:2003.00982 (2020).


**Conclusion:**

While the proposed method is sound and interesting and the paper has some good evaluations. The main problem of the paper is represented by the unclear theoretical part that is detrimental to the paper and should be fixed. In this form, I tend to give a *5: marginally below*  rating.

__________________________________________________________
**After rebuttal**

I thank the authors for their responses, but I will keep my original rating of the paper. I think this paper has potential and a new submission will have a better result.

Suggestion for improvements:

1. As the other reviewers pointed out, the proposed method is computationally expensive. Although this is a downside, it is not really a major one, as long as the authors have a proper analysis of the complexity and of the actual inference / training time. The authors pointed out that their contribution consists of proving that the adaptive receptive fields are useful, and I agree that an efficient method is not necessary for this, but the analysis should be made.

2. I think that additional ablation studies are needed. For example, the experiment in Figure 4 a) supports Proposition 1 but only on Cora that is homophilic and where the mean of the nodes is a good aggregation. The experiments with added noise are a good start. Maybe also compare GRARF with GAT with a simple strategy of selecting the edges (top k according to $e_{i,j}$, or all with $\alpha_{i,j} > 0.5$ ).

3. Improve the motivation from Section 2.

---

> ### Author Response · Authors · 2020-11-23
> **Official reply to Rev.4**
>
> We thank you for your insightful comments on the theoretical part of the paper. We understand that your major concern is **why small weights lead to bad representations**.
>
> The reason why small, soft weights suffer is that the central node cannot *concentrate* on important contexts in the neighborhood, i.e. a limited number of nodes cannot have dominant influence on the central node. As a result, the representations of nodes with high degrees is largely determined by its common neighbors, instead of potentially important contexts.
>
> An intuitive example is to consider how the variation of neighbor's features effects the representation of the central node: consider $h_i = \sum_{j \in N(v_i)} \alpha_{ij} W h_j$ (ignoring the activation), and a variation of features on $v_k$, i.e. $\tilde{h_k}= h_k + \epsilon$, then $\Delta h_i = \alpha_{ik} W \epsilon=O(\frac1n)$. That is, the effect of one or several contexts on the central node is always limited, even if they are totally removed ($\Delta h_i = \tilde{\alpha_{ik}} h_i - \alpha_{ik}Wh_{k}=O(\frac1n)$, where $\tilde{\alpha_{ik}}=\frac{e_{ik}}{Z-e_{ik}}=O(\frac1n)$). This harms the network to model potential *dominant effect* of important contexts.
>
> Also, another contribution of our paper in showing the drawbacks of soft attention weights are the experimental results in Fig. 3. These results show that the soft weights are not only small, but also condense over $\frac1n$ (especially considering the 75% interval). This further hinders the model's ability in distinguishing important neighbors.
>
> We would also like to stress that the problem is *essential* on graphs compared with grid-like data, since in grid-like data such as natural languages and images, the neighborhood size is always limited (e.g. 8 neighboring pixels in images); while in graphs, such limitation is not available.

---

> > ### Comment · AnonReviewer4 · 2020-11-24
> > **Reply**
> >
> > I thank the authors for their response.
> >
> > I agree that small weights make it impossible for the model to concentrate on important nodes. This is an important aspect but I suggest that the authors emphasize the cases when this would be crucial. For example, for homophilic datasets like Cora a good model is not required to focus on certain nodes. This is shown by the good performance of simple models like SGC.
> >
> > It would also be important to give the exact inference time for GRARF as stated in the initial review and also all the methods in Table 2.

---

### Official Review · AnonReviewer1 · 2020-10-25
**Official Blind Review #1**

**Rating:** 5
**Confidence:** 2

**Review:**

Summary:
The authors theoretically and empirically show that soft-attention mechanism uses in GCNs suffers from over-smoothness in large neighborhoods. For addressing this shortcoming, they propose a neighborhood sampling approach called adaptive receptive fields (ARFs) which discretely select nodes among the multi-hop neighborhood and allow to efficiently explore long-distance dependencies in graphs. The authors also propose GRARF (GCN with Reinforced ARF) which learns optimal policy of constructing ARFs with reinforcement learning. For a given node, an RL agent successively expands ARF via a two-stage process. Firstly, a contact node in an intermediately-constructed ARF is selected and then a context among the direct neighbors of the contact node is added to ARF. The reward is the performance of the trained GCN on constructed ARF. Overall, the results demonstrate the effectiveness of the approach on benchmark datasets. The authors also demonstrate that the method is quite effective at handling noise in the graph compared to GCN and GAT by evaluating them on the Cora dataset with synthetically added noise.

Strengths:
1. The proposed idea of using reinforcement learning for selecting influence neighborhood of a node in GCNs is novel. The results show the effectiveness of the method.
2. The method is more robust to noise and is capable of capturing information from distant neighbors for learning efficient representation.

Weaknesses:
1. The scalability of the method for larger graphs is not very explicit from the given experiments.
2. Using RL might accompany multiple challenges involved with learning policy which might restrict the applicability of the approach.


Questions:
1. In Figure 4(b)-(d), it is not clear which attention weight distribution corresponds to GCN and which corresponds to the GAT model.
2. Please explain the reason behind the vast difference in the performance of the baseline models on cora, citeseer, and pubmed datasets. For instance, GAT reported around 83% on cora, however, here it has been given as 87.8. Also, on ppi dataset, the performance changes from 97.3 to 47.29.

-----
Based on the issues pointed out by the other reviewers. I have decided to reduce my score for the paper.

---

> ### Author Response · Authors · 2020-11-21
> **Official reply to Rev.1**
>
> We truly appreciate your recommendation for our paper. With regard to your questions, below are some explanations:
>
> 1. All Figure 4 (b) - (d) show the distributions of attention weights in GAT, while the distributions are conditioned on **different degrees of nodes** (d=2,5,10).
>
> 2. The differences of reported accuracies are caused by different evaluation setups:
>
> a) Different dataset splits are used in GAT [1] and in our experiments on Cora: in GAT, 20 samples from each class are extracted as the training set, while in our paper, we use a fixed ratio of training set instead.
>
> b) GATs in [1] use a vast model with 3 layers & 1024 hidden dims (in total) to achieve high performances on PPI, while all baselines uses 2 layers & 128 dims in our paper.
>
> For more implementation details, you may refer to Section A in our Appendix.
>
> 3. About the weakness of leveraging RL:
>
> Using RL to construct ARFs is indeed an expediency. Nevertheless, we believe the main contribution of our paper is to point out the necessity of constructing ARFs via discrete node selection (regardless of the very approach), and the efficiency of GRARF, especially considering inferencing, is actually competent. The complexity of the action space of GRARF is $O(T+d)$, and the inference complexity (including the constructing of ARFs) is $O(T \times (T + d))$. When $T = O(d)$, this is the same as all two-layer GCNs with neighborhood aggregation (most typically, GraphSAGE).

---

### Official Review · AnonReviewer3 · 2020-10-26
**Recommendation to weak reject**

**Rating:** 5
**Confidence:** 4

**Review:**

#####Summary#####

This paper proposes to construct adaptive receptive fields for graph convolution using reinforcement learning. This strategy can address the smoothness of attention weights issue, and capture long-distance dependencies in graphs. Considering the following pros and cons comprehensively, I decide to give a recommendation of weak reject to this work.

#####Pros#####

(1) The studied problem is very crucial and valuable. Aggregating information from neighborhood can capture useful information as well as introduce noise sometimes. How to construct an adaptive receptive field of each node is a significant direction that deserves more investigation.

(2) This work points out the issue that the soft attention weights (such as in GAT) suffer from over-smoothness in large neighborhoods. Also, the impressive experimental result in Fig. 3 can verify this to some degree. This observation is insightful and might helpful for future exploration.

#####Cons#####

(1) The main shortcoming of this work is the high complexity of the method. This work utilizes the reinforcement learning to training the constructor, which is expected to be very complicated. Furthermore, there does not exist the experiments about running efficiency of the proposed method, which is very necessary to demonstrate the value of the proposed approach.

(2) Also, the dataset used in this paper is relatively small and some of them are shown to have data quality issue [1]. More experiments on larger datasets, such as OGB [1], should be considered.

[1] Hu et al. Open graph benchmark: Datasets for machine learning on graphs.

---

> ### Author Response · Authors · 2020-11-23
> **Official reply to Rev.3**
>
> We thank you for your review comments. To address your concerns, we would like to clarify:
>
> 1. **Complexity of RL**
>
> Indeed, incorporating RL is an expendiency, as extra complexity may thus be introduced. Nonetheless, we believe the major contribution of our paper is to show the drawback of soft node weights and thus the necessity of a discrete context selection process (regardless of the very approach). Meanwhile, although high complexity is a common first impression of RL, the  efficiency of GRARF, especially considering inferencing, is actually competent. The (averaged) complexity of the action space of GRARF is $O(T+d)$, and the inference complexity (including the constructing of ARFs) is $O(T \times (T+d))$. When $T=O(d)$ (e.g. $T=10$ in our implementation), this complexity is *the same as all two-layer GCNs* with neighborhood aggregation (most typically, GraphSAGE).
>
> 2. **The choice of datasets**
>
> We appreciate your suggestions on moving forward from conventional baselines including cora et al to OGB, and we'll certainly consider using OGB in our future work. However, we would also like to clarify that currently, Cora & Citeseer are still (almost the most) widely used benchmarks in GNN papers including GraphSAGE(2017), GAT(2018), MixHop(2019), DropEdge(2020). We believe the most important standard in evaluating the effectiveness of experiments is whether there is fairness between baselines, and we made it sure through our experimental setups. Also, we think the scales of Github & PPI datasets are large enough to varify the effectiveness of GRARF.

---

> > ### Comment · AnonReviewer3 · 2020-11-23
> > **Response to authors' reply**
> >
> > Thank you for your response about my concerns.
> >
> > 1. Could you provide the comparison of the running time between CRARF and baselines? It would be more convincing to show the empirical computational cost. This point is very necessary to demonstrate the efficiency of the proposed method.
> >
> > 2. It would be more convincing if more larger datasets are included in the future. Overall, I agree that the considered datasets are enough currently.

---

### Official Review · AnonReviewer2 · 2020-10-26
**my review**

**Rating:** 5
**Confidence:** 5

**Review:**

To address the over-smoothness in neighborhood caused by soft-attention in GNN, this paper presents an idea of adaptive receptive fields (ARFs), which can choose contexts on different hops from the central node, so as to efficiently explore dependencies with longer distances.  The construction of ARFs follows a reinforcement learning (RL) framework.

The adaptive selection of contextual neighbors is a promising solution, as not all the neighbors are equally important. It is an interesting idea to apply reinforcement learning to this problem. An RL agent first selects  a contact node in the direct neighbors, and then explores the neighbors of this contact node.

Although the design of ARF with RL is a valuable practice,  the ARF construction process has several issues. It defines the state of a target node as a vector from f_s by taking the searched neighbors as input. How f_s is defined? The action is to select a neighbor of a target node u from its 1-hop or 2-hop neighbors. Even when the states transit from one to another, the action is always to select neighbors of u from its 1-hop or 2-hop neighbors, rather than going deep to explore the graph for more useful neighbors. After a fixed number of steps T, the added neighbors are still within the 2-nd order neighbors. Such an ARF construction process limits the region to explore in the graph. It cannot "explore dependencies with longer distances".

The presented approach shares a lot similarity with that in [1]. The constructor is analogous to the score network f_s, while the evaluator is analogous to the network f_h and f_c for accumulating the neighbor with recurrent attentions and for classification. However, [1] is not cited, and not used as a baseline to compare.

In addition, the over-smoothing problem for GNN has been recently studied in several papers, authors are suggested to cite and compare with them as well, such as [2] [3].

Suggestions to the authors:
1) The design of ARF should be improved, with a reasonable setting of the state function and actions.
2) Recent work should be aware. Discussion and experimental comparison with them should be included.

[1] Akujuobi et al. Recurrent Attention Walk for Semi-supervised Classification. WSDM 2020

[2] Chen et al. Measuring and relieving the over-smoothing problem for graph neural networks from the topological view. AAAI 2020.

[3] Xie et al. When Do GNNs Work: Understanding and Improving Neighborhood Aggregation. IJCAI 2020


Thanks to authors for the clarification.  I have updated my score.
However, a easy search with the title of the paper "Recurrent Attention Walk for Semi-supervised Classification" in Google can find the code in GitHub.

---

> ### Author Response · Authors · 2020-11-21
> **Official reply to Rev.2**
>
> We thank you for all the suggestions you proposed, while we think the decision and some comments are based on misunderstanding of our paper. Accordingly, we would like to clarify:
>
> 1. "How f_s is defined?"
>
> We put the concrete definition of $f_s(N_a(u))$ in *Section 4.1: GRARF implementation*, that is,
> $$
> f_s(N_a(u)) = W_s[h_u^c||\frac1n\sum_v h_v^c] + b_s.
> $$
> We are sorry to cause the confusion.
>
> 2. "...the action is always to select neighbors of u from its 1-hop or 2-hop neighbors..."
>
> This is not true. The *contact node*, $a_t^1$, is chosen from all nodes in the ARF $N^{(t)}_a(u)$ instead of the 1-hop neighborhood $N^1(u)$. Indeed, in the first two steps, selected nodes inevidably lie in 1- or 2-hop neighborhoods of $u$; while later on, further (say 3-hop) nodes may be included in the ARF by choosing 2-hop nodes as contact nodes. Also, the distributions of *dependency length* (i.e. #hops of ARF nodes) are demonstrated in Fig.3 in the Appendix, where long dependencies (>2 hop) are frequently observed.
>
> We sincerely hope that the reviewer may consider re-rating for our paper given that there are major misunderstandings.
>
> 3. Missing citation of [1].
>
> We thank you for bring citation [1] in site, as it does share a similar motivation with GRARF. However, we notice that the datasets it used do not overlap with ours, and there is no available implementation of the paper. Especially when RL is incorporated, comparing the results of [1] and ours are factually difficult. Meanwhile, we find that the walk process in [1] is similar to a *depth-first exploration* of the neighborhood, while GRARF with 2-stage action spaces is more analogous to a *breadth-first exploration* (actually more flexible than BFS). When it comes to constructing ARFs, we would anticipate that the BFS-like exploration is more reasonable. In fact, we find that [1] can be regarded an instance of GRARF, where $a_t^1$ is constrained to be the most newly added node in the ARF.
>
> 4. Missing citation of [2] [3]
>
> We would like to again emphasize (footnote 3 in our paper) that the smoothness -- *weight smoothness* -- discussed in our paper is essentially different to the smoothness more oftenly discussed -- *feature smoothness* (discussed in [2] [3], which originates from [4] cited in footnote 3). As we mainly focus on *weight smoothness* problem in our paper, the discussion of *feature smoothness* is thus brief due to the limitation of space.
>
> [4] Li et al. Deeper insights into graph convolutional networks for semi-supervised learning, ICML 2018.

---

### Decision · Program_Chairs · 2021-01-07
**Final Decision**

**Decision:**

Reject

**Comment:**

Four knowledgeable referees lean towards rejection because of the missing detailed complexity analysis [R1,R2,R3], the choice of rather small datasets which hinders the rigorous evaluation of GNN models [R3,R4], missing state-of-the-art comparisons [R2] and ablations [R4]. The rebuttal addressed some of the concerns raised by the reviewers, in particular, clarifications request by R2, smoothness of the weights questions of R4, and the difference in performance of the baseline methods of R1. However, after discussion, the reviewers are still concerned with the missing ablations, comparisons, and complexity analysis. I agree with their assessment and therefore must reject. However, I agree with the reviewers that this is an interesting approach and encourage the authors to consider the reviewer's suggestions for future iterations of their work.